# The Multi-Strain Probiotic OMNi-BiOTiC^®^ Active Reduces the Duration of Acute Upper Respiratory Disease in Older People: A Double-Blind, Randomised, Controlled Clinical Trial

**DOI:** 10.3390/microorganisms11071760

**Published:** 2023-07-05

**Authors:** Maja Strauss, Dušanka Mičetić Turk, Mateja Lorber, Maja Šikić Pogačar, Anton Koželj, Ksenija Tušek Bunc, Sabina Fijan

**Affiliations:** 1Faculty of Health Sciences, University of Maribor, Žitna ulica 15, 2000 Maribor, Slovenia; 2Faculty of Medicine, University of Maribor, Taborska ulica 8, 2000 Maribor, Slovenia; 3Community Healthcare Center dr. Adolf Drolc Maribor, 2000 Maribor, Slovenia

**Keywords:** upper respiratory tract infection, URTI, duration of illness, probiotics, multi-strain, older people, immune function

## Abstract

Immunosenescence is the adverse change in the human immune function during aging, leaving older people more prone to an increased risk of infections and morbidity. Acute upper respiratory tract infections (URTIs) are very common among older people, often resulting in continued morbidity and mortality. Therefore, approaches, such as consuming probiotics, that shorten the duration or even reduce the incidence of URTIs in older people are being studied. The aim of this study was to determine the effects of a multi-strain probiotic OMNi-BiOTiC^®^ Active, which contains 11 live probiotic strains, on the incidence, duration, and severity of URTIs in older people. In this randomized double-blinded placebo-controlled study, 95 participants, with an average age of 70.9 years in the probiotic group and 69.6 years in the placebo group, were randomly allocated to two groups: 10^10^ cfu per day of the multi-strain probiotic intervention OMNi-BiOTiC^®^ Active (49) or placebo (46). The incidence of URTIs in older people after 12 weeks supplementation with OMNi-BiOTiC^®^ showed no statistically significant difference between the two groups (*p* = 0.5244). However, the duration of the URTI infections was statistically significantly different between the groups (*p* = 0.011). The participants that consumed the probiotic had an average duration of illness of 3.1 ± 1.6 days, whilst participants that received the placebo had symptoms for an average of 6.0 ± 3.8 days (*p* = 0.011). Statistically significant differences in lymphocyte counts in both groups after supplementation (*p* = 0.035 for the probiotic group and *p* = 0.029 for the placebo group) and between both groups were found (*p* = 0.009). Statistically significant differences in eosinophil (*p* = 0.002) and basophil counts (*p* = 0.001) in the probiotic groups before and after supplementation with probiotics were also found. Supplementation with the multi-strain probiotic OMNi-BiOTiC^®^ Active may benefit older people with URTIs. Larger randomised controlled clinical trials are warranted. Clinical Trial Registration; identifier NCT05879393.

## 1. Introduction

The average global population is ageing due to a decline in fertility and an increase in life expectancy [1]. According to the United Nations’ World Population Prospects, the number of older people aged 60 or over will most likely double by 2050 [2]. Although life expectancy has been increasing in recent decades, many functions decline during ageing; for example, a more frequent onset of chronic diseases. Immunosenescence is the adverse change of the human immune function with ageing and includes a multifactorial and dynamic phenomenon defined as the gradual deterioration of the immune system with aging, involving alterations to the immune organs, immune cells, and immune-related molecules that affect both natural and acquired immunity and play a critical role in most chronic diseases, and potentially leads to an increased risk of infections and certain cancers in the older people [3,4,5,6,7,8,9,10,11]. Sometimes, the term inflammaging is used with regard to older people and that triggers an anti-inflammatory response to counteract the age-related proinflammatory environment [12]. In addition to the above mentioned immunosenescence and inflammaging, the gut microbiome undergoes significant changes with aging in the diversity and loss of resilience of the intestinal microbiota, which can lead to permissive communication along the gut microbiota–lung axis [6,7,13,14,15,16]. 

Colds, influenza, and infections of the throat, nose, sinuses, and upper airways are collectively called acute upper respiratory infections (URTIs). Acute URTIs usually get better within three to seven days [4,5]. Common viral pathogens that cause URTIs are influenza viruses, respiratory syncytial viruses, parainfluenza viruses, rhinoviruses and enteroviruses, adenoviruses, coronaviruses, and others, whilst common bacterial pathogens causing URTIs are *Streptococcus pneumoniae, Streptococcus pyogenes, Haemophilus influenzae, Moraxella catarrhalis, Mycoplasma pneumoniae,* and others [6,17,18,19,20,21,22]. URTIs are common among nursing home residents, representing about one-third of all infections, and they are the most frequent reason for hospital admittance and a significant cause of mortality [6,23,24,25]. 

Probiotics are defined as “live microorganisms that, when administered in adequate amounts, confer a health benefit on the host” [26]. Probiotics are most commonly specific strains of several bacterial or fungal genera, including: *Lacticaseibacillus, Lacticaseibacillus, Lactiplantibacillus*, *Limosilactobacillus*, *Lactobacillus, Lactobacillus Bifidobacterium*, *Lactococcus,* and *Saccharomyces* [27]. Based on clinical studies, utilising strains of these species can achieve a statistically significant health benefit in the probiotic group compared to the placebo group [28]. Probiotics are well known to influence various gut microbiota–organ axes, including the gut microbiota–lung axis via ‘crosstalk’ [29,30]. A recent systematic review [6] found that certain probiotic strains such as *Lacticaseibacillus paracasei* subsp. *paracasei* CNCM I-1518, *Lacticaseibacillus paracasei* Shirota, *Lactobacillus delbrueckii* subsp. *bulgaricus* OLL1073R-1, *Loigolactobacillus coryniformis* K8 CECT5711, and *Bacillus subtilis* CU1 were more effective than a placebo in either reducing the incidence, duration, or symptoms of URTIs in older people [8,9,31,32,33,34]. Other clinical studies utilising similar probiotic strains such as *Lacticaseibacillus paracasei* Shirota and *Lacticaseibacillus rhamnosus* GG [35,36] found no differences in URTI incidence or duration among older people. All these clinical trials with older people utilised single-strain probiotic strains. Another study also confirmed that a specific multi-strain probiotic formulation with similar strains reduced the incidence of URTIs among athletes [37]. 

The aim of this clinical trial was to assess the effect of the multi-strain probiotic OMNi-BiOTiC^®^ Active, which contains 11 live probiotic strains (10^10^ cfu per day), on the incidence, duration, and severity of URTIs in older people. 

## 2. Materials and Methods

### 2.1. Trial Design

The study was a randomised, double-blind, placebo controlled clinical trial including participants from two winter seasons: one in the winter season of 2020/2021 and the other in the winter season of 2021/2022. Participants were assigned sequentially in a 1:1 ratio either to the intervention (probiotics) or the control (placebo) group. The randomisation procedure is explained in Section 2.8. The clinical trial was registered retrospectively in the US National Library of Medicine (ClinicalTrials.gov) under number NCT05879393 and was approved by the relevant regional ethics committees. The reporting of the clinical trial followed the CONSORT 2010 statement [38,39,40]. 

### 2.2. Participants

Healthy independent-living adults aged ≥65 years were recruited through local retirement associations in the Slovenian Styria region where co-authors MS, DMT, KTB, or SF briefed the participants at information meetings. The categorization of the age of older people being 65 years and over is in line with the definition of the OECD indicator [41] and with the age of eligibility for old-age pension in Slovenia [42]. All participants willing to participate had given their written consent. Participation was free and voluntary. 

The participants were included in the clinical trial if they were 65 years or older at the signing of the written consent form. Other inclusion criteria were the ability of the participant to eat independently and to adhere to all procedures of the clinical study (daily preparation of drink, completing the questionnaires, and recording symptoms of respiratory tract infections). The participants were excluded if they experienced an exacerbation of an existing chronic disease, an exacerbation of metabolic diseases, mental incapacity to understand instructions, and if they were prescribed long-term usage of antibiotics. Participants were also excluded if they changed their eating habits or consumed any probiotics 14 days before inclusion in the clinical trial. Blood samples before and after consuming probiotics/placebo were drawn from the participants in the first arm (winter season of 2020–2021). Participants with markedly abnormal results of blood tests were excluded. 

The nutritional assessment of the participants was based on the full version of the mini nutritional assessment questionnaire (MNA), which is a validated screening tool for health care of older people [43,44]. According to this tool, a person is considered malnourished if the malnutrition indicator score of MNA is less than 17 points. The person is at risk of malnutrition if it is between 17 and 23.5 points, and a person is considered well nourished if it is ≥24 points. The participants completed this assessment before recruitment. Only well-nourished participants were included in the trial. 

All participants received a questionnaire for upper respiratory tract infections (URTI questionnaire), adapted from several clinical trials [45,46,47], and kept a daily record during the onset of URTI symptoms. A detailed explanation of the questionnaire is noted Section 2.4. 

All participants were asked to maintain their normal eating habits and refrain from consuming other probiotics. If a participant received antibiotics, they were asked to specifically note this in the URTI questionnaire. The participants received a brochure prepared by MS, DMT, and SF with general information on the aim of the clinical trial, information on the probiotics and placebo used, instructions for preparing the drink, and con-tact information. After completing the clinical trial, all participants were notified whether they belonged to the probiotic or placebo group.

### 2.3. Intervention

Participants were randomised to receive either the probiotic or placebo. Participants in the probiotic group received the multi-strain probiotic OMNi-BiOTiC^®^ Active as a lyophilised powder, where 2 g contained a total of 5 × 10^9^ cfu of the following live probiotic strains: *Lacticaseibacillus casei* W56 (DSM 26388), *Lactobacillus acidophilus* W37 (DSM 26412), *Ligilactobacillus salivarius* W24 (DSM 26403), *Levilactobacillus brevis* W63 (DSM 26393), *Lactococcus lactis* W58 (DSM 26390), *Lactococcus lactis* W19 (DSM 26399), *Bifidobacterium animalis* subsp. *lactis* W52 (DSM 26334), *Bifidobacterium longum* subsp. *longum* W108 (DSM 26405), *Bifidobacterium breve* W25 (DSM 26332), *Bifidobacterium animalis* subsp. *lactis* W51 (DSM 26333), and *Bifidobacterium bifidum* W23 (DSM 26331). The probiotic and placebo were consumed twice daily, thus the daily cfu of probiotic strains was 10^10^ cfu per day. 

The additional ingredients included: rice starch, maltodextrin, plant protein, potassium chloride, magnesium sulphate, and manganese sulphate. The participants in the placebo group received a similar looking and tasting powder without the probiotic strains and plant protein. Both powders contained no animal protein, gluten, yeast, or lactose. Nutritional information was as follows per 2 g (= single dose): 7.46 kcal, fat 0.02 g (of which saturates < 0.01 g), carbohydrates 1.69 g (of which sugars < 0.01 g), protein 0.14 g, and salt < 0.01 g. 

Both the probiotic product and the placebo were identical in appearance, smell, and taste and were provided by the Institute AllergoSan, GmbH, Graz, Austria, in indistinguishable containers labelled with consecutive numbers. The participants prepared their probiotic drink twice daily (in the morning and in the evening) by suspending 2 g of the lyophilised powder in 1–2 dL of water and waiting for 1 min to allow the probiotics to re-activate before consuming the drink. The study included a 12-week consumption phase and a two-month follow-up phase without consumption. 

### 2.4. Reporting of Respiratory Tract Infections

Respiratory tract infections were defined as the occurrence of at least one of the symptoms noted in the URTI questionnaire, adapted from several clinical trials [45,46,47]. Seven symptoms, including nasal (stuffed nose, secretion, or sneezing), conjunctivitis (red eyes), pharyngeal (sore throat or secretion), and bronchial symptoms (cough or secretion), as well as headache, myalgia, and fever (oral temperature > 37.7 °C) were assessed. The participants rated their symptoms using a four-level scale where 0 = no symptoms, MI = mild symptoms, MO = moderate symptoms, and SE = severe symptoms on the first day of the onset of symptoms and on the third day and noted the date of the onset of symptoms and the date of the cessation of symptoms. In the case of a URTI episode, the participants recorded their symptoms and the duration of the URTI episode. 

The URTI questionnaire contained evaluation tables for three separate upper respiratory tract infection episodes. If more than three episodes occurred a new questionnaire was sent. Two evaluators (MS and SF) regularly contacted the participants by mobile phone and e-mail asking them about symptoms and possible side effects and reminding them to fill in their URTI questionnaire in the case of respiratory tract infection episodes. The URTI episodes were reported during all 12 weeks of treatment and until the completion of the 2-month follow up.

### 2.5. Biological Analysis 

Venous blood from the group of participants in the first winter season between 2020 and 2021 was collected by qualified nurses (20 mL) in the morning. One sample was collected at the beginning of the 12-week intervention period and another at the end of this period. The blood analysis included parameters related to immunological functions such as the concentration of blood cells: leukocytes (lymphocytes, monocytes, eosinophils, basophils, and neutrophils) and immunoglobulin IgA [3,48]. 

### 2.6. Study Outcomes

The primary outcome was the comparison of the incidence of acute upper respiratory tract infections during the 12 weeks of the intervention in the whole population of both arms. The secondary outcome was the comparison of the duration of acute upper respiratory tract infections during the 12 weeks of the study in the whole population of both arms. Additional secondary outcomes were the changes in the series of blood indictors during the intervention of the participants in the first winter season related to immunological functions. 

### 2.7. Sample Size

A sample size calculation was performed based on published data. According to de Vrese and co-authors [49], we presumed that the incidence of URTIs would be experienced by 60% of our participants. We also presumed that the incidence of URTI symptoms would be approximately 48% lower in the probiotic group compared to the placebo group, as noted by Fonollá and co-authors [33]. The minimal number of participants was then calculated using the online Clin Calc tool (ClinCalc LLC) for the sample size for two independent groups (probiotic and placebo) with a dichotomous endpoint. We set the statistical parameters as follows: 5% alpha level, 80% power, and enrolment ratio 1:1. Assuming a dropout of 20%, a total of 94 participants were to be recruited.

### 2.8. Randomisation

Randomisation into two groups (probiotic and placebo) was performed with blocks of four on a 1:1 ratio using the computer program: Open-Source Research Randomizer Tool. Participants were allocated to the probiotic group or to the control group using an individual randomisation number (study product allocation concealed) and were included sequentially in accordance with the randomisation list. Randomisation was conducted by the supplier who did not participate in the clinical study as an investigator. The randomisation list was opened after the last participant completed their follow-up period and after all URTI questionnaires were returned. 

### 2.9. Blinding

Both the multi-strain probiotic and placebo were identical in appearance, taste, nutritional composition, and packaging to assure blinding of participants, investigators, and outcome assessors. Statistical analyses were completed by M.S., S.F., and M.Š.P.

### 2.10. Statistical Analysis

Analyses were performed according to intention-to-treat principle in which all patients were included. The baseline characteristics and data of the MNA questionnaire were collected upon recruitment. The blood analysis data of the group in the first winter season of 2020 and 2021 were obtained at the beginning of the intervention and after the 12-week intervention. The data from the URTI questionnaires were collected at the end of the follow-up period. Statistical analyses were performed with IBM Statistical Package for Social Sciences (IBM SPSS 28.0, SPSS Inc., Chicago, IL, USA). 

Descriptive statistics (means and standard deviations) were generated for continuous variables, and the number of patients and percentages are given for discrete variables of the baseline characteristics. Comparisons of the age, body mass index, and the MNA score between the probiotic and placebo groups were conducted using Student’s t-tests for independent samples, whilst the comparisons of the percentage of male to female ratio and the percentage of chronic non-communicable diseases were conducted using Fisher’s exact test. Blood analysis was also reported using descriptive statistics (mean and standard deviation) and the comparison of the blood parameters of both the probiotic and the placebo groups was conducted using a paired t-test. The comparison of the incidence of URTI symptoms between both the probiotic and the placebo group (primary outcome) was assessed using Fisher’s exact test, whilst the comparison of the duration of URTI symptoms and the severity of URTI symptoms between both the probiotic and the placebo groups (secondary outcome) were conducted using the Mann–Whitney U test. The significance level for all comparisons was set at *p* < 0.05. 

## 3. Results

As shown in Table 1, a total of 95 independently living participants were recruited and randomly distributed into two groups: 49 in the probiotic group and 46 in the control group (placebo), respectively. The clinical study took place in two winter seasons: 59 participants in the winter season between November 2020 and March 2021, and 36 participants in the winter season of November 2021 to March 2022. All 95 participants completed the 12-week intervention period and the two-month follow-up period. Analysis of blood indicators was conducted for the subset of 59 participants in the first winter season (32 in the probiotic group and 27 in the placebo group, respectively). Thus, the final analysis included 95 participants (49 in the probiotic group and 46 in the placebo group, respectively). 

Table 1 shows that no significant differences were detected in any of the baseline characteristics of the participants. Of the 95 study participants, 49 (51.6%) received the probiotics and 46 (48.4%) received the placebo. The probiotic group consisted of 17 (34.7%) men and 32 (65.3%) women; the placebo group consisted of 15 (32.6%) men and 31 (67.4%) women. The mean age was 70.9 ± 5.1 years in the probiotic group and 69.9 ± 4.3 years in the placebo group. Half of the participants (*n* = 23; 50.0%) in the placebo group reported the presence of chronic non-communicable diseases, while in the probiotic group, 23 (46.9%) reported the presence of chronic non-communicable diseases. In both groups, participants had an average body mass index over 25 kg/m^2^. There were no differences between the groups regarding the mini nutritional assessment score (*p* = 0.088). 

The study flowchart is depicted in Figure 1 and followed the CONSORT 2010 participant flowchart [39].

During the 12-week treatment period and 2-month follow-up period, a total of 32 URTI episodes were reported by the study participants; of these, 15 were in the probiotic group, whilst 17 were in the placebo group (Table 2). More than one URTI event per person was reported by two participants in the placebo group. Although the incidence of infection was lower in the probiotic group (30.61%) compared to the placebo group, (36.96%) no statistically significant difference was found between the two groups (*p* = 0.524). The duration of the URTI infection was in fact statistically significantly different between the placebo and test groups (*p* = 0.011), where participants that consumed the probiotic (N = 15) had an average duration of illness of 3.1 ± 1.6 days, whilst the participants that received the placebo (N = 14) who developed URTI infections had symptoms for an average of 6.0 ± 3.8 days. 

The effect of probiotic supplementation on the severity of individual URTI symptoms (nasal, pharyngeal, bronchial, headache, myalgia, and fever) among the 32 URTI episodes are depicted in Figure 2. The severity of individual symptoms were rated as mild (1), moderate (2) and severe (3). Although it is obvious from Figure 2 that the average severity of all reported symptoms was lower in the probiotic group compared to the placebo group, none of the differences were statistically significant.

The changes in the blood parameters related to immunological functions for the participants of the first winter season are noted in Table 3. 

Table 2 shows that the leucocyte count decreased slightly in the probiotic group, whilst it increased slightly in the placebo group, but neither change was statistically significant (*p* > 0.05). Similar results were found for the monocyte count. The segmented neutrophil count remained unchanged in both the placebo and the probiotic group. The lymphocyte count statistically significantly decreased in the probiotic group (*p* = 0.035), whilst in the placebo group the lymphocyte count statistically significantly increased (*p* = 0.029). Comparing both groups, a statistically significant difference in the lymphocyte count was found after supplementation (*p* = 0.009). The eosinophil and basophil counts statistically significantly decreased in the probiotic group, whilst in the placebo group the eosinophil and basophil counts decreased, but this was not statistically significant (*p* > 0.05). No statistically significant differences were found for IgA in either the probiotic or the placebo group. 

## 4. Discussion

This randomised placebo-controlled clinical trial with 95 participants, aged 65 years or older, assessed the efficiency of using the multi-strain probiotic OMNi-BiOTiC^®^ Active, which contains five lactobacilli and bifidobacterial strains as well as two lactococcal strains (10^10^ cfu per day), for the reduction in the incidence and duration of upper respiratory tract infections (URTIs) for two winter seasons. The URTI symptoms were recorded using a validated self-assessment-based questionnaire [45,46,47]. Our results report that the duration of URTI symptoms was statistically significantly lower in the probiotic group compared to the placebo group (*p* = 0.017), as the average duration of URTI symptoms was 3.1 ± 1.6 days in the probiotic group, whilst the participants that received placebo had URTI symptoms for an average of 6.0 ± 3.8 days. Although the overall incidence of URTI infections in our study was lower in the probiotic group (30.61%) compared to the placebo group (36.96%), the difference was not statistically significant (*p* > 0.05). 

Three other clinical trials had similar results to our study [31,32,49]. Two of these clinical trials included older people only and utilised probiotic strains of *Lacticaseibacillus paracasei* at different concentrations in fermented drinks. In the clinical trial by Turchet and co-authors [31], older people consumed a fermented dairy drink containing the probiotic *Lacticaseibacillus paracasei* subsp. *Paracasei* CNCM I-1518 (10^8^ cfu/mL), whilst in the clinical trial by Fujita and co-authors [32], older people consumed a fermented milk containing a different probiotic strain *Lacticaseibacillus paracasei* Shirota (4 × 10^10^ cfu). The third clinical trial, with results published in two articles [46,49], included adults and older people (aged between 18 and 67) that consumed a multi-strain probiotic supplement containing *Lactobacillus gasseri* PA 16/8, *Bifidobacterium longum* SP 07/3, and *Bifidobacterium bifidum* MF 20/5 (5 × 10^7^ cfu). On the other hand, some clinical studies have shown different results; for example, [3] one found that daily supplementation of *Lacticaseibacillus paracasei* N1115 (3.6 × 10^7^ cfu) in adults/elderly people over 45 years resulted in a statistically significant lower incidence of URTI, but not duration of URTI. Guillemard and co-authors [8] found that supplementation with *Lacticaseibacillus paracasei* subsp. *paracasei* CNCM I-1518 (10^10^ cfu) statistically significantly reduced the incidence and duration of URTI episodes among older people. A statistically significant lower incidence of URTIs, or other winter infections among older people, was also found after daily consumption of *Lactobacillus delbrueckii* subsp. *bulgaricus* OLL1073R-1 [9], and after daily consumption with *Bacillus subtilis* CU1 (2 × 10^9^ cfu) [34]. No statistically significant differences for the incidence or duration of URTI episodes among older people after daily consumption with *Lacticaseibacillus paracasei* Shirota (6.5 × 10^9^ cfu) were reported in the clinical study by Van Puyenbroeck and co-authors [35]. Two additional clinical studies that investigated the incidence of URTI episodes, but not their duration [33,36], also found no statistically significant differences for the incidence of URTI episodes among older people after the consumption of *Lacticaseibacillus rhamnosus* GG (10^9^ cfu) [36] or *Loigolactobacillus coryniformis* K8 CECT_5711_ (3 × 10^9^ cfu) [33]. As was found in a recent review on the efficiency of probiotics for reducing the incidence or duration of upper respiratory tract infections in older people [6], probiotic strain selection is one of the important steps in preparing a well-designed clinical trial. Another recent Cochrane review [5] found that probiotics may reduce the number of participants diagnosed with URTIs. Our study utilised multiple-strain probiotics and found statistically significant health benefits in the probiotic group compared to the placebo group. Multi-strain probiotics can synergistically employ different mechanisms, including managing infectious diseases, inhibiting antibiotic-resistant pathogens, overall health improvement, and gut microbiota immunity and modulation [27,50,51]. A recent in vitro study [52] that also utilised the multi-strain probiotic OMNi-BiOTiC^®^ Active found that multi-strain probiotics exhibited higher antimicrobial effect than single-strain probiotics, perhaps due to synergistic action.

The severity of all seven investigated URTI symptoms in our study did not result in statistically significant differences between the groups; however, all seven symptoms were reportedly lower in severity in the probiotic group compared to the placebo group. Similarly, the severity of URTI symptoms were lower but not statistically significant in the probiotic group compared to the control group in the clinical trial by Pu and co-authors [3]. Similarly, in the clinical study by Hor and co-authors [53], where both adults and older people consumed *Lacticaseibacillus casei* Zhang, some URTI symptoms were statistically significantly lower in duration compared to the placebo for the adult participants, but not for the participants older than 65 years old. The authors concluded that the adult population benefited more than the elderly population, mainly due to the better reception of the adult host towards *Lacticaseibacillus casei* Zhang, and they postulated that the better reception could be due to the probiotic’s ability to decrease the measured proinflammatory (IL-1) cytokines and increase the measured anti-inflammatory cytokines (Il-4, Il-10) in the plasma of the adults after probiotic consumption. In the clinical study by de Vrese and co-authors [46,49] that included adults and older people, they found statistically significant less severe symptoms for the typical common cold symptoms but not for flu-like symptoms in the probiotic group compared to the placebo group. 

The results of changes in immune cells and immune-related molecules in blood serum before and after treatment with probiotics were assessed in the subset of participants, that were recruited in the first winter season (2020–2021) just before one of the lockdowns that occurred in Slovenia due to the COVID-19 pandemic. It is obvious from the scientific literature that perceived stress can affect the immune status, especially in older people, and that the communication between the gut microbiota and the brain can modulate the host immune system [54,55]. Another important factor to consider is that alterations in the number and composition of lymphocytes, and that their subsets in blood are a hallmark of immune system aging [56]. In our study, we did not find any statistically significant changes in the average concentrations of the immune-related molecule immunoglobulin A (IgA) or in the average concentration of several immune cells after probiotic consumption in the probiotic group compared to the placebo group. Most other clinical studies on adults or older people have also reported only slight differences or no statistically significant differences in most immune-associated cells or molecules in the average concentrations of measured immunological parameters [8,33,34,46,49]. In our study, the reductions in the average concentration of eosinophils, basophils, and lymphocytes before and after probiotic consumption were statistically significant in the probiotic group compared to placebo; although, all average concentrations were within the normal ranges. We also found a statistically significant difference in the lymphocyte count between both groups, which indicates a reduction in inflammation in the probiotic group. A reduction in these parameters in the probiotic group could perhaps be explained by the more successful action of the immune system in the participants that consumed probiotics. Different cell lines, including monocytes, have been shown to influence the release of proinflammatory and anti-inflammatory cytokines in response to certain probiotic bacteria [46,49] and their transfer into tissues to clear bacterial, protozoa, fungal, and viral infections [57]. Other immunological parameters that decrease inflammation, such natural killer cells and the cytokine interferon gamma produced by natural killer cells and interleukin-10 (Il-10), which are critical for innate and adaptive immunity against pathogens, can significantly increase after probiotic consumption [9,34]. However, these parameters were not measured in our study. Probiotics have also been found to decrease the concentration of inflammatory cytokines such as tumour necrosis factor alpha (TNF-alpha) and interleukin-1 (Il-1) in older people [53,58]. The duration of probiotic consumption could also present a factor in the clinical study by Hor and co-authors [53]; it was found that a 12-month consumption of *Lacticaseibacillus casei* Zang reduced plasma proinflammatory (IL-1) and increased anti-inflammatory cytokines (IL-4, IL-10) compared to the placebo (*p* < 0.05), whilst in our study, our participants consumed the multi-strain probiotic for 12 weeks.

Strain selection of probiotics is one of the important factors [53], as different strains of the same species can vary in their ability to exhibit health benefits to the host, and employ different mechanisms and targets. However, it seems that in older people, other factors, such as immunosenescence, play an equally important role in the effect of probiotics [7,8,9]. Probiotics can, however, have other important health benefits for older people such as reducing constipation, which increases with age and can result in gut dysbiosis, which shifts the pool of gut metabolites leading to delayed mobility. Probiotics can also indirectly influence the immune system through modulating the gut microbiota and improve cognitive functions [53,59]. 

Our research had limitations. Part of the clinical study took place during COVID-19 epidemic, with related preventative measures used. This means that the older people who participated in the clinical study had a lower risk of contact with pathogens that cause upper respiratory tract infections. However, despite this limitation, a statistically significant lower duration of URTIs and certain blood parameters related to immunological function were found in the probiotic group compared to the placebo group. 

## 5. Conclusions

Overall, we found that probiotics significantly reduced the duration of acute URTIs in older people. Additional randomised, controlled clinical studies that focus on strain selection, duration of probiotic consumption, and the influence of contributing factors of this population, such as influenza vaccination, immunosenescence, and malnutrition, and considering living accommodations that differentiate between independent-living participants and participants living in old-age care homes should be conducted.

## Figures and Tables

**Figure 1 microorganisms-11-01760-f001:**
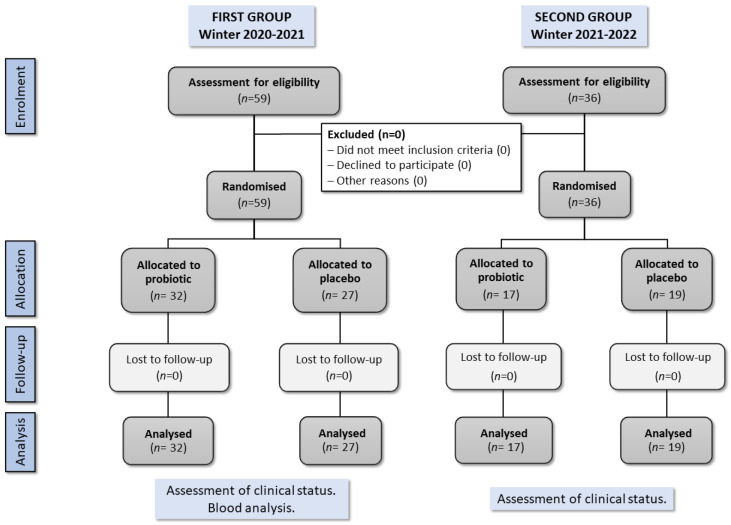
Modified CONSORT 2010 participant flowchart of the randomised controlled trial (RCT) investigating the effect of OMNi-BiOTiC^®^ Active on the symptoms of upper respiratory tract infections among older people.

**Figure 2 microorganisms-11-01760-f002:**
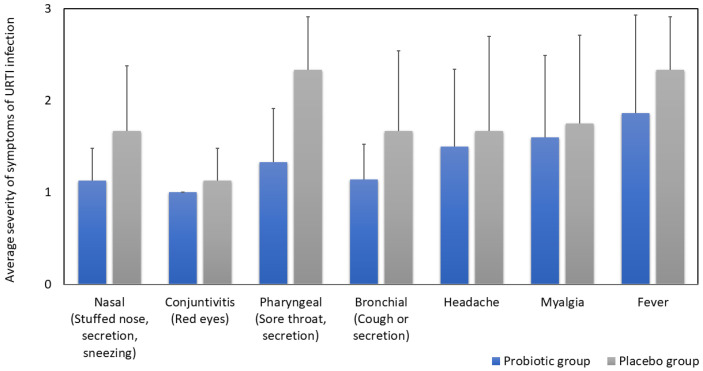
The effect of OMNi-BiOTiC^®^ Active on the average severity of individual symptoms among the 32 events of upper respiratory tract infections in older people. Values are mean with SD, Mann–Whitney U test (*p* > 0.05) for all samples.

**Table 1 microorganisms-11-01760-t001:** Baseline characteristics of study participants (*n* = 95).

Characteristic	Probiotic Group (*n* = 49)		Placebo Group (*n* = 46)		*p* Value
	Mean or %	SD	Mean or %	SD	
Male to female ratio					
Number (M/F)	17/32		15/31		
% (M/F)	34.7/65.3		32.6/67.4		0.581 *
Age (years)	70.9	5.1	69.6	4.3	0.082 ^#^
Body mass index (kg/m^2^)	26.4	3.7	26.9	4.3	0.285 ^#^
Chronic non-communicable diseases					
Yes/No (%)	46.9/53.1		50.0/50.0		0.441 *
Mini Nutritional Assessment form	27.0	2.6	25.8	4.0	0.088 ^#^

Legend: SD—standard deviation; * Fisher’s exact test; ^#^ Student’s t test for independent samples; *p* < 0.05.

**Table 2 microorganisms-11-01760-t002:** Effect of OMNi-BiOTiC^®^ Active on the incidence and duration of upper respiratory tract infections among older people.

Variables of URTI Events	Probiotic Group (*n* = 49)		Placebo Group (*n* = 46)		*p*-Value
Total number of URTI events, number (%)	15	30.61%	17	36.96%	*p* = 0.524 *
Number of participants with URTI events	15	30.61%	14	30.43%	
0 URTI events per participant, number (%)	34	69.39%	32	69.57%	
1 URTI event, number of participants (%)	15	30.61%	12	26.09%	
2 URTI events, number of participants (%)	0	0%	1	2.17%	
3 URTI events, number of participants (%)	0	0%	1	2.17%	
Duration of URTI episodes, mean (SD)	3.1	(1.6)	6.0	(3.8)	*p* = 0.011 **

* Fisher’s exact test, ** Mann–Whitney U test, SD—standard deviation, URTI—upper respiratory tract infection.

**Table 3 microorganisms-11-01760-t003:** Changes in the immune cells and immune-related molecules in blood samples of 59 participants before and after treatment with probiotics or placebo.

Blood Assessment		Probiotic Group (N = 32)				*p* Value ^#^	Placebo Group (N = 27)				*p* Value ^#^
Parameter	Unit	Baseline		12 Weeks			Baseline		12 Weeks		
		Mean		SD			Mean		SD		
Leukocytes	10^9^/L	6.2	1.5	6.0	1.4	0.169	6.8	1.6	7.0	2.1	0.329
Neutrophils segm.	10^9^/L	3.3	1.1	3.3	1.1	0.889	3.8	1.2	3.8	1.5	1.000
Lymphocytes	10^9^/L	2.2	0.8	1.9	0.6	**0.035 ***	2.2	0.6	2.6	1.0	**0.029 ***
Monocytes	10^9^/L	0.54	0.14	0.50	0.15	0.113	0.55	0.14	0.57	0.16	0.227
Eosinophils	10^9^/L	0.25	0.15	0.16	0.08	**0.002 ***	0.20	0.11	0.19	0.12	0.852
Basophils	10^9^/L	0.06	0.05	0.02	0.04	**0.001 ***	0.04	0.05	0.03	0.04	0.327
IgA	g/L	2.6	1.2	2.4	1.0	0.267	2.0	0.8	2.1	1.0	0.344

^#^ paired *t*-test (* *p* < 0.05); SD—standard deviation.

## Data Availability

No new data other than the published data were created.

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
