# Peer review of "The Multi-Strain Probiotic OMNi-BiOTiC® Active Reduces the Duration of Acute Upper Respiratory Disease in Older People: A Double-Blind, Randomised, Controlled Clinical Trial"

_microorganisms, 2023, doi:10.3390/microorganisms11071760_

Round 1
Reviewer 1 Report
Meritorious trial. Omnibiotic Active vs placebo: better in reduction of disease length in elderly with respiratory infections. Introductions is too long. References: check the format. Language light polishing needed.
see ahead
Author Response
Thank you for all your remarks and suggestions. They have helped to improve the manuscript.

Reviewer 2 Report
below are notes I took as I went through the study.
But here are the major points
-it was registered after study was started. That is a serious issue. Not sure how the editors want to handle it. Maybe there are some internal docs, IRB, etc that show the primary outcome, etc did not change.
-writing needs work. That is relatively easy. But more importantly, just needs a lot of cutting. Why do we need such a long intro and discussion. This is a great study. A well thought out RCT. It essentially speaks for itself don't let that get lost in too many words.
Abstract
severe course of disease, could just say morbidity
For sure should list all the strains but not sure you need that in an abstract
Don’t tell the reader what test you did in the abstract, “Mann-Whitney U test”
Did you not do a stat test for this as no mention in the abstract, “The participants that consumed the probiotic had an average duration of illness of 3.1 ± 31 1.6 days, whilst participants that received placebo had symptoms for an average of 6.0 ± 3.8 days.”
INTRO
Not sure people are getting older because of the “decline in fertility” You may mean average age but even that is not relevant.
The intro can be a lot shorter and writing needs tightened. Cannot say things like On the other hand too informal.
The reader knows that the elderly are at an increased risk of infectious disease and poorer outcomes.
Don’t need to do things like list this, “: Lacticaseibacillus rhamnosus, 84
Lacticaseibacillus casei, Lactiplantibacillus plantarum, Limosilactobacillus reuteri, Lactobacillus 85
acidophilus, Lactobacillus delbrueckii subsp. bulgaricus, Lactobacillus gasseri, Bifidobacterium in- 86
fantis, Bifidobacterium longum, Lactococcus lactis, Enterococcus faecium, Streptococcus ther- 87
mophilus, Bacillus subtilis, Escherichia coli, Saccharomyces cerevisiae var. boulardii and others” It doesn’t help the reader.
Do you know the CFUs for each strain? That would be interesting.
METHODS
Up to the editors but the trial was registered retrospectively. Ie incorrectly.
Wonder why they didn’t use the Wisconsin URI scale but maybe the ones you used are just as good
https://www.fammed.wisc.edu/wurss/
RESULTS
Are those the only demographics you have for Table 1
In results you don’t need to tell us about methods, “Mann-Whitney U”
To me Table 2 is pretty boring, but Table 3 much more interesting, as is Figure 2, I would switch that around.
DISCUSSION
Need to work on the writing but think you have what you need. But could cut a lot.
I am confused why your primary outcome was incidence when you point out very nicely that most studies have found no change in incidence but change in duration?
Not sure I buy this, “Literature often concludes that multistrain probiotics are more efficient as 428
each strain can synergistically employ different mechanisms, including managing infec- 429
tious diseases, inhibiting antibiotic-resistant pathogens, overall health improvement, gut 430
microbiota immune and modulation” When one looks at the evidence base, ie well done RCTs, most are not multistrain probiotics.
I am not sure what you are talking about in line 433
see above. But needs rewritten. Need to cut and work on grammar.
Author Response

(The authors gave the same response as above.)

Reviewer 3 Report
The manuscript is very well written. The research was well designed. Ethical permission has been granted. Limitations of the study are also highlighted.
The authors may or may not accept the following suggestions (the paper is really readable and well thought out):
- Explain why you have significantly more women than men in your study.
- Emphasize that while participants were monitored for not taking other probiotics in pill or powder form, it was not monitored for participants consuming fermented foods that may contain probiotics
- Did the participants, given that they were older, forget to drink the probiotic and report it? How did you level the scores in that case.
-Reference needs to be sorted. They should be numbered in the text.
-Considering that the same probiotic was tested, you can discuss this work as well:
Fjan, S., Kocbek, P., Steyer, A., Vodičar, P.M. and Strauss, M., 2022. The Antimicrobial Effect of Various Single-Strain and Multi-Strain Probiotics, Dietary Supplements or Other Beneficial Microbes against Common Clinical Wound Pathogens. Microorganisms, 10(12), p.2518.
Author Response
Thank you for all your remarks and suggestions. They have helped to imporve the manuscript.
